# Robust Integral Sliding Mode Control for Uncertain Impulsive Stochastic Systems with Time Delays

*Note: Sub-titles are not captured in Xplore and should not be used

Lei Song
line 2: *dept. name of organization
(of Affiliation)*
line 3: *name of organization (of
Affiliation)*
Chengdu, China
songlei1372022@163.com

*Abstract*—**This paper studies the integral sliding mode control (ISMC) for uncertain impulsive stochastic systems with time delays. An ISMC law is proposed by designing a new integral sliding surface. Then, by constructing a piecewise time-dependent Lyapunov function, the uniformly almost surely exponentially stability conditions of the closed-loop system are established. Furthermore, an optimal design algorithm of solving control gains is formulated based on the established stable conditions and linear matrix inequalities (LMIs) theory.**

*Keywords—uncertain impulsive stochastic systems, integral sliding surface, solving control gains*

## I. INTRODUCTION

The impulse system is used to describe the phenomenon of abrupt variation of state that occurs in many practical fields, such as orbital transfer of satellite, sampled-data control systems (Benford, 2008; Naghshtabrizi, Hespanha, & Teel, 2008; Pfeiffer & Foerg, 2005), etc. Thus, the control design of the impulsive systems becomes an important and challenging task in recent years and has attracted widespread attention in control field, for example see Briat (2013), Briat and Seuret (2012a, 2012b), Chen, Ruan, and Zheng (2017), Chen, Zhang and Lu (2021), Chen and Zheng (2009) and Li, Peng, and Cao (2020). Among them, sufficient stability conditions were established in Briat and Seuret (2012a, 2012b) for linear impulsive systems by utilizing a class of looped-functional Lyapunov functions, while the authors in Briat (2013) investigated the stability analysis problem of linear aperiodic impulsive systems through the use of time-varying discontinuous Lyapunov functions. The Lyapunov stability for impulsive systems is investigated in Li et al. (2020) by using event-triggered impulsive control. The authors in Chen and Zheng (2009) studied the problems of robust stability and $H\infty$-control for uncertain impulsive systems with time delay and (Chen et al., 2017; Chen, Zhang et al., 2021) investigated the $L2$-gain or $L2 \times l2$-gain control design problems for linear impulsive time-delay systems.

## II. SYSTEM DESCRIPTION AND PRELIMINARIES

Consider the linear uncertain impulsive stochastic systems with time delay:

$$\begin{cases} dz(t) = [A_i(t)z(t) + A_{di}(t)z(t-\tau) + B_i(u(t) + g(t, z(t), z(t-\tau)))]dt \\ \quad + C[D_i(t)z(t) + D_{di}(t)z(t-\tau)]dw(t), \ t \neq t_v \\ z(t_v) = E_i z(t_v^-), \ v \in N \\ z(t_0 + \zeta) = \varphi(\zeta), \ \zeta \in [-\tau, 0] \end{cases}$$

(1)

where $u(t) \in R^m$ is the control input, $z(t) \in R^n$ is the system state, $g(t, z(t), z(t-\tau)) : R_+ \times R^n \times R^n \to R^m$ is the nonlinear uncertainty. In the systems $A_i(t) = A_i + \Delta A_i(t)$, $A_{di}(t) = A_{di} + \Delta A_{di}(t)$, $D_i(t) = D_i + \Delta D_i(t)$, $D_{di}(t) = D_{di} + \Delta D_{di}(t)$, $A_i \in R^{n \times n}$, $A_{di} \in R^{n \times n}$, $C_i \in R^{n \times m}$, $D_i \in R^{m \times n}$, $D_{di} \in R^{m \times n}$, $E_i \in R^{n \times n}$ are real constant matrices, $B_i \in R^{n \times m}$ is the control input matrix, $\text{rank}(B_i) = m$ and $\text{rank}(E_i) = n$. $\Delta A_i(t)$, $\Delta A_{di}(t)$ and $\Delta D_i(t)$, $\Delta D_{di}(t)$ are the parameter uncertainties and uncertainties of stochastic perturbation, respectively. $\{t_v\}_{v \in \mathcal{N}_0} \in \mathcal{T}(\sigma_0, \sigma_1)$ is the impulse time sequence, $w(t)$ is a one-dimensional Wiener motion on the complete probability space $(\Omega, \mathcal{F}, \mathcal{F}_t, P)$.

Assumption 1[]. There exist two matrices of proper dimensions $S_{1i}$, $S_{2i}$, $T_{1i}$, $T_{2i}$ such that $\begin{bmatrix} \Delta A_i(t) & \Delta A_{di}(t) \\ \Delta D_i(t) & \Delta D_{di}(t) \end{bmatrix} = \begin{bmatrix} S_{1i} \\ S_{2i} \end{bmatrix} O_i(t) \begin{bmatrix} T_{1i} & T_{2i} \end{bmatrix}$, and the unknown matrix function $O_i(t)$ satisfies $\|O_i(t)\| \leq 1$.

Assumption 2 []. There are known constants $\alpha_1$, $\alpha_2$ such that nonlinear uncertain function $g(t,z(t),z(t-\tau))$ satisfies $\|g(t,z(t),z(t-\tau))\| \le \alpha_1\|z(t)\|+\alpha_2\|z(t-\tau)\|$.

Definition 1 The zero solution of system (1) is uniformly mean-square exponentially stable over $\mathcal{T}(\sigma_0,\sigma_1)$, if there exist two positive scalars $X$ and $\alpha$ such that

$$E\|z(t)\|^2 \le XE\|\varphi\|_\tau^2 e^{-\alpha(t-t_0)}, \quad t \ge t_0$$

for any initial data $\varphi \in PC_{\mathcal{F}_{t_0}}^b([-\tau,0];R^n)$.

Lemma 1 []: Given $0 < A \in R^{n\times n}, B \in R^{m\times n}, C \in R^{p\times n}, D_1 \in R^{n\times q}, D_2 \in R^{m\times q}, 0 < E \in R^{m\times m}$. If there exists a constant $\varepsilon > 0$ such that

$$\begin{bmatrix} -A & B^T \\ * & -E^{-1} \end{bmatrix} + \varepsilon^{-1}\bar{D}\bar{D}^T + \varepsilon\bar{C}^T\bar{C} < 0 \tag{2}$$

where $\bar{D}^T = \begin{bmatrix} D_1^T & D_2^T \end{bmatrix}$ and $\bar{C}^T = \begin{bmatrix} C & 0_{p\times m} \end{bmatrix}$, then the following inequality holds

$$-A + He(D_1GC) + (B+D_2GC)^T E(B+D_2GC) < 0 \tag{3}$$

for any matrix $G \in R^{q\times p}$ satisfying $\|G\| \le 1$.

The control objective in this study is to construct a SMC strategy to guarantee that the uncertain stochastic linear impulsive system with time delay (1) is uniformly mean-square exponentially stable.

### III. Integral Sliding Mode Control Design

#### A. Sliding Mode Surface Design

It is follows from the ISMC theory that the ISMC law is chosen the following form

$$u(t) = u_1(t) + u_2(t) \tag{4}$$

where the role of $u_1(t)$ is stabilizing the dynamic and $u_1(t)$ is designed to reject the disturbance and maintains the dynamic. To design the sliding mode surface, a continuous sliding mode function which depend on the impulse information is proposed at first. Then, $u_1(t)$ and $u_2(t)$ is designed to assure the reachability of the sliding mode surface.

Inspired by [] and [], we design an integral sliding surface for system (1) as:

$$s(t) = Qx(t) + \upsilon(t)Q_1 x_p(t) - \int_0^t [Q(Ax(\varsigma) + A_d x(\varsigma-\tau) + Bu_1(\varsigma)) - \frac{1}{\sigma_0}Q_1 x_p(\varsigma)]d\varsigma \tag{5}$$

where $Q$ is designed to satisfy that $QB$ is nonsingular and $QC = 0$, $Q_1 = Q(I-J)$, and the control law $u_1(t)$ is chosen as $u_1(t) = Kx(t) + K_d x(t-\tau)$, in which $K$ and $K_d$ are selected such that $A + BK$ and $A_d + BK_d$ are Hurwitz matrices, respectively. Owing to that $QB$ is nonsingular and $\text{rank}(B) = m$, there exists a matrix $Z > 0$ such that $Q = B^T Z$.

Because the states $x(t)$ is discontinuous at $t_\kappa$, it can be seeming from the continuous of $s(t)$ at $t_\kappa$ that the sliding surface $s(t)$ established for system (1) is continuous on $[0,\infty)$. Based on $\upsilon(t_\kappa) = 1$ and $\upsilon(t_\kappa^-) = 0$, it follows from $QC + Q_1 = Q$ that

$$s(t_\kappa) = (QC + Q_1)x(t_\kappa^-) - \int_0^{t_\kappa} Q(A+BK)x(\tau)d\tau$$
$$= s(t_\kappa^-) \tag{6}$$

so, $s(t)$ is continuous on $[0,\infty)$.

To guarantee the reachability of the designed sliding surface $s(t) = 0$, the SMC law $u_2(t)$ is designed in Theorem 1.

Theorem 1: Consider the system (1) under Assumptions -, if we adopt the sliding mode function () and the SMC law:

$$u_2(t) = -\phi(t)\text{sign}(s(t)) \tag{7}$$

where

$$\phi(t) = \phi_0 + \|(QB)^{-1}QS_1\|(\|T_1 x(t)\| + \|T_2 x(t-\tau)\|)$$
$$+ \alpha_1\|x(t)\| + \alpha_2\|x(t-\tau)\|$$

with $\phi_0 > 0$ being a scalar, then the reachability of the sliding mode surface $s(t) = 0$ can be ensured.

Proof: For $t \in [t_\kappa, t_{\kappa+1}), \kappa \in \mathcal{N}_0$, the time derivation of $s(t)$ along (8) as follows:

$$\dot{s}(t) = Q[\Delta A(t)x(t) + \Delta A_d(t)x(t-\tau) + B(u(t) + g(t,x(t),x(t-\tau)))] - QBu_1(t) \tag{8}$$

Substituting (10) into (11), we get

$$\dot{s}(t) = Q[\Delta A(t)x(t) + \Delta A_d(t)x(t-\tau) + B(-\phi(t)\text{sign}(s(t)) + g(t,x(t),x(t-\tau)))] \tag{9}$$

Choosing the Lyapunov function $V = \frac{1}{2}s^T(t)(QB)^{-1}s(t)$. From (12), we have

$$\dot{V}_1 = s^T(t)(QB)^{-1}Q(\Delta A(t)x(t)+\Delta A_d(t)x(t-\tau)) + s^T(t)(u_2(t)+g(t,x(t),x(t-\tau)))$$
$$\leq \|s(t)\|\|(QB)^{-1}QS_1\|(\|T_1 x(t)\|+\|T_2 x(t-\tau)\|)+s^T(t)u_2(t)+\|s(t)\|(\alpha_m\|x(t)\|+...$$
$$\leq -\phi_0\|s(t)\|$$
$$\leq -\phi_0\phi_1(V_1(t))^{1/2}$$
(10)

where $\phi_1 = (2\lambda_{\min}(QB))^{1/2}$. Since $s(t)$ is continuous on $t \in [0,\infty)$, we obtain $V(t)=0$, $\forall t \geq \tilde{T} \triangleq (2/(v_0 v_1))(V(0^+))^{1/2}$, which means that $s(t)=0$ for $\forall t \geq \tilde{T}$.

*B. Stability of sliding motion*

Once $t \geq \tilde{T}$, we get $s(t)=\dot{s}(t)=0$ from Theorem 1. By solving $\dot{s}(t)=0$, the equivalent control law $u_{equ}(t)$ for $u_2(t)$ can be designed as follows:

$$u_{equ}(t) = -(GB)^{-1}G[\Delta A(t)x(t)+\Delta A_d(t)x(t-\tau)]-g(t,x(t),x(t-\tau))$$
(11)

Substituting $u(t)=u_1(t)+u_{equ}(t)$ into (1) establishes the following systems

$$\begin{cases} dx(t) = [(A+BK+(I_n-\bar{B})\Delta A(t))x(t)+(A_d+BK_d+(I_n-\bar{B})\Delta A_d(t))x(t-\tau)]dt \\ \quad + C[D(t)x(t)+D_d(t)x(t-\tau)]dw(t), \quad t \neq t_\kappa \\ x(t_\kappa)=Jx(t_\kappa^-), \kappa \in N \\ x(t_0+\zeta)=\varphi(\zeta), \zeta \in [-\tau,0] \end{cases}$$
(12)

where $\bar{B}=B(B^T ZB)^{-1}B^T Z$.

To set up the criterion of stability for system (15), the interval $[t_\kappa,t_{\kappa+1})$ is divided into $N$ subintervals $\Delta_{\kappa i} \triangleq [t_\kappa+id_\kappa, t_\kappa+(i+1)d_\kappa), i \in \overline{0,N-1}$, in which $d_\kappa = \dfrac{t_{\kappa+1}-t_\kappa}{N}$. Then, we define:

$$\theta_{0i}(t) = \frac{t_\kappa+(i+1)d_\kappa-t}{d_\kappa}, t \in \Delta_{\kappa i}, i=\overline{0,N-1}, \kappa \in \mathcal{N}_0$$
(13)

Let $\theta_{1i}(t)=1-\theta_{0i}(t)$. For $\{t_\kappa\}_{\kappa \in \mathcal{N}_0} \in \mathcal{T}(\sigma_0,\sigma_1)$, there is $m_0(t):R_+ \to [0,1]$ satisfying

$$\bar{\upsilon}(t) = \frac{m_0(t)}{\sigma_0}+\frac{m_1(t)}{\sigma_1}$$
(14)

where
$$m_1(t)=1-m_0(t)$$
,

$$\bar{\upsilon}(t) = \begin{cases} \dfrac{\upsilon(t)-1/\sigma_1}{d_2\|\alpha_0(t-1\tau)\phi_1}, & if \ \sigma_0 \neq \sigma_1 \\ 1, & if \ \sigma_0 = \sigma_1 \end{cases}$$

To exploit the impulsive structure of system (15), the piecewise time-dependent Lyapunov function candidate is introduced

$$V_2(t,x(t)) = \varsigma(t)x^T(t)P(t)x(t)+\int_{t-\tau}^{t}e^{-r(t-s-\tau)}x^T(s)Rx(s)ds$$
(15)

In (), $\varsigma(t)=\mu^{\upsilon(t)}$, $P(t)=\sum_{i=0}^{N-1}P_i(t)\tilde{\chi}_i(t)$,

$P_i(t)=\sum_{j=0}^{1}\theta_{ji}(t)P_{i+j}$, $t \in \Delta_{\kappa i}$, $P_{\tilde{i}}>0$, $\tilde{i} \in \overline{0,N}$, $\tilde{\chi}_i(t)$ is the characteristic function. It is obviously that $\tilde{V}(t)$ is continuous in each impulse time intervals $(t_\kappa,t_{\kappa+1})$, $\kappa \in \mathcal{N}_0$.

Based on the fact that $0 \leq \upsilon(t) \leq 1$, there is a number $\tilde{\upsilon}_0(t) \in [0,1]$ such that

$$\varsigma(t) = \sum_{l=0}^{1}\tilde{\upsilon}_l(t)u^{1-l}$$
(16)

where $\tilde{\upsilon}_1(t)=1-\tilde{\upsilon}_0(t)$.

Theorem 2: Consider the sliding mode dynamics (15), for a given positive number $\mu$, $c$, if there exist matrices $0 < P_{\tilde{i}} \in R^{n \times n}, \tilde{i} \in \overline{0,N}$, $0 < Z \in R^{n \times n}$ and scalars $\varepsilon_{ij}>0$, $i \in \overline{0,N}$, $j \in \overline{0,1}$, such that :

$$\begin{bmatrix} \Psi_{ikl} & 0 \\ 0 & -Z^{-1} \end{bmatrix}+\varepsilon_{ij}^{-1}\bar{E}\bar{E}^T+\varepsilon_{ij}\bar{H}\bar{H}^T < 0$$
(17)

$$J^T P_0 J \leq \mu P_N$$
(18)

$$B^T ZD = 0$$
(19)

where

$$\bar{E}^T = [S_1^T P_{i+k}I_1 \ S_1^T], \quad I_1=[I_n \ 0_n], \quad \bar{H}^T=[\bar{T} \ 0],$$

$$\bar{T}=[T_1 \ T_2], \quad \Psi_{ikjl}=\begin{bmatrix} \Theta_{ikjl} & P_{i+k}(A_d+BK_d) \\ * & -u^{l-1}R \end{bmatrix}+\Delta_{ik}$$

,then system (15) is uniformly mean-square exponentially stable over $\mathcal{T}(\sigma_0,\sigma_1)$.