# OpenReview forum: "Robust Integral Sliding Mode Control for Uncertain Impulsive Stochastic Systems with Time Delays"
_IEEE.org/ICIST/2024/Conference — IEEE ICIST 2024 Conference Submission_

### Official Review · Reviewer_NQdq · 2024-08-20
**Recommended rejection**

**Rating:** 3
**Confidence:** 4

**Review:**

The innovation points of this manuscript are insufficient and the logic is unclear. It is not recommended to publish this paper in ICIST 2024.

---

### Official Review · Reviewer_JQe4 · 2024-08-21
**Recommended rejection**

**Rating:** 3
**Confidence:** 3

**Review:**

The paper lacks a significant contribution to the existing body of knowledge and fails to provide a clear advancement or novel insights into integral sliding mode control for stochastic systems. As such, I do not think this paper is suitable for ICIST 2024.

---

### Official Review · Reviewer_YDts · 2024-08-21
**This manuscript contains numerous formatting errors, presents simplistic content, lacks innovation, and appears to be an unfinished version of a paper. Therefore, it is recommended for rejection.**

**Rating:** 3
**Confidence:** 3

**Review:**

This manuscript contains numerous formatting errors, presents simplistic content, lacks innovation, and appears to be an unfinished version of a paper. Therefore, it is recommended for rejection. Some of the major issues are listed below:
1. Formulas are excessively long and overlap, such as formula (5).
2. The paper lacks experimental and simulation validations.
3. The introduction is overly cursory.
4. The paper is deficient in necessary content such as references.

---

### Decision · Program_Chairs · 2024-09-08

Reject